# Michael Acceptor Compounds as Hemoglobin Oxygen Affinity Modulators for Reversing Sickling of Red Blood Cells

**DOI:** 10.3390/ph18060783

**Published:** 2025-05-24

**Authors:** Khadijah A. Mohammad, Asala H. Naghi, Mohini S. Ghatge, Benita Balogun, Mariana Macias, Salma Roland, Albert Opare, Osheiza Abdulmalik, Martin K. Safo, Abdelsattar M. Omar, Moustafa E. El-Araby

**Affiliations:** 1Department of Pharmaceutical Chemistry, Faculty of Pharmacy, King Abdulaziz University, Jeddah 21589, Saudi Arabia; kmohammad@kau.edu.sa (K.A.M.); asala.naghi@gmail.com (A.H.N.); asmansour@kau.edu.sa (A.M.O.); 2Department of Medicinal Chemistry, School of Pharmacy and Center for Drug Discovery, Virginia Commonwealth University, Richmond, VA 23219, USA; msghatge@vcu.edu (M.S.G.); oparea@vcu.edu (A.O.); dawoudme@vcu.edu (M.E.E.-A.); 3Division of Hematology, The Children’s Hospital of Philadelphia, Philadelphia, PA 19104, USA; benitab@sas.upenn.edu (B.B.); mariana7@sas.upenn.edu (M.M.); rsal@sas.upenn.edu (S.R.); abdulmalik@email.chop.edu (O.A.); 4Department of Pharmaceutical Organic Chemistry, Faculty of Pharmacy, Helwan University, Ain Helwan, Cairo 11795, Egypt

**Keywords:** hemoglobin, sickle cell disease, Michael acceptor, antisickling, Hb oxygen affinity, polymerization

## Abstract

**Background/Objectives**: Sickle cell disease (SCD) is caused by a β-globin gene mutation (βGlu6Val) that produces sickle hemoglobin (HbS). When deoxygenated, HbS polymerizes, leading to red blood cell (RBC) sickling; therefore, hemoglobin is a central therapeutic target for SCD. Current strategies include increasing the levels of oxygenated HbS (which cannot polymerize) and/or directly destabilizing the deoxygenated HbS polymer. This study aimed to design and synthesize next-generation Michael acceptor antisickling hemoglobin modifiers (MMA-206, MMA-207, MMA-208, and MMA-209) and evaluate their antisickling efficacy. **Methods:** Four Michael acceptor compounds (MMA-206 to MMA-209) were synthesized and characterized. Their pharmacologic activities and modes of action were assessed in vitro using disulfide exchange reaction with normal hemoglobin, sickling inhibition assays with sickle red blood cells, and hemoglobin oxygen equilibrium curve analysis with normal and sickle red blood cells. **Results:** MMA-206 exhibited the strongest antisickling activity, outperforming previously studied Michael acceptor antisickling agents. All four MMA analogues bound to hemoglobin at βCys93, destabilizing the low-oxygen-affinity T-state and thereby preventing deoxygenation-induced HbS polymerization and RBC sickling. In addition, they appeared to directly destabilize the HbS polymer, indicating a second mechanism of action. Furthermore, time-dependent oxygen equilibrium measurements confirmed that their pharmacologic effect was sustained over time in vitro. **Conclusions:** The new Michael acceptor compounds, particularly MMA-206, demonstrated potent antisickling effects via dual mechanisms and showed sustained activity. These findings highlight Michael acceptor compounds’ promise as hemoglobin oxygen-affinity modulators for the treatment of SCD.

## 1. Introduction

Sickle cell disease (SCD) is the most common inherited hematologic disorder worldwide, affecting over 20 million people [1,2,3]. The disease is caused by a single point mutation in the β-globin gene of hemoglobin (Hb), replacing βGlu6 in normal hemoglobin (HbA) with βVal6, resulting in sickle hemoglobin (HbS) [4,5,6,7]. When deoxygenated, HbS polymerizes, causing red blood cells (RBCs) to become rigid and sickle. This leads to intravascular hemolysis of RBCs that triggers ischemia–reperfusion, associated with NO deficiency, hypertension, inflammation, oxidative stress, vaso-occlusive crisis (VOC), progressive organ damage, and, eventually, premature death [4,5,6,7,8,9,10]. Given Hb’s central role in the pathophysiology of the disease, it has become a key target for drug discovery efforts to treat SCD [4]. One validated pharmacologic approach is the use of aromatic aldehydes, which prevent RBC sickling by increasing the affinity of Hb for oxygen to increase the concentration of the non-polymer-forming oxygenated HbS (oxy-HbS) [4,11,12,13,14,15,16,17,18,19]. Aromatic aldehydes exhibit this pharmacologic property by forming reversible Schiff-base interactions with the N-terminal αVal1 nitrogen of Hb α-subunits and, through additional protein interactions, stabilizing the high-O_2_ affinity R-state Hb [4,11,12,13,14,15,16,17,18,19]. Several aromatic aldehydes have been studied for their potential for the treatment of SCD, with one such compound, Voxelotor, approved in 2019 for the treatment of SCD [4,13,14,15,16,19,20,21]. However, Voxelotor was withdrawn from the market in 2024, presumably due to modest efficacy and a poor risk–benefit profile. With very few exceptions (excluding Voxelotor), a major drawback of aromatic aldehydes, which has hampered their development as therapeutic agents, has been their poor pharmacokinetic (PK) properties due to significant and rapid metabolic oxidation of the aldehyde moiety into the corresponding inactive metabolites by aldehyde dehydrogenase (ALDH), particularly in the liver and RBCs [4]. Both the chronic nature of SCD, as well as the large amount of intracellular Hb in the blood (millimolar concentrations) that needs to be modified to achieve a therapeutic effect, clearly require potent and long-acting antisickling agents. Hence, efforts have been or are being made to replace the aldehyde group with different metabolically stable electrophiles, including those that form irreversible covalent interactions [4,22,23,24,25].

To overcome the poor pharmacokinetic limitations of aromatic aldehydes, our group rationally designed a new class of antisickling compounds incorporating a stable Michael acceptor reactive center [4,22]. We hypothesized that the compounds would form an irreversible covalent interaction with βCys93 on the surface of the β-subunit of Hb, inhibiting a salt-bridge interaction between βAsp94 and βHis146. Inhibition of this salt bridge formation stabilizes the R-state and destabilizes the T-state of Hb, resulting in the increased affinity of Hb for oxygen [4,22]. The Michael acceptor concept was based on previously reported studies of ethacrynic acid (ECA; Figure 1), which binds to βCys93 to increase Hb oxygen affinity [26]. Although these earlier “KAUS” Michael acceptor compounds were metabolically stable, they did not exhibit significant antisickling activity and, in some instances, even decreased Hb affinity for oxygen [4,22]. Unexpectedly, crystallographic study of a representative compound, KAUS-15 (Figure 1), showed that those compounds bind at the α-cleft of deoxygenated Hb to make covalent interactions with the N-terminal αVal1 amines of the two α-subunits. The carboxylate group on the compound seems to direct the molecules to the α-cleft of deoxygenated Hb instead of binding to βCys93, stabilizing the T-state and reducing Hb affinity for oxygen, which is counter to the desired pharmacologic effect [4,22].

Removal of the carboxylate moieties in the KAUS compounds expectedly directed the next generation azolylacryloyl derivatives (exemplified by KAUS-38 in Figure 1) to bind to βCys93 [4,23]. As expected, these compounds increased the affinity of Hb for oxygen, leading to effective sickling inhibition. Also, as anticipated, the compounds exhibited long duration of action in vitro [4,23]. Nonetheless, the antisickling potencies of the KAUS compounds were weak to moderate compared to those of aromatic, aldehyde-based antisickling agents, and they showed limited solubility [4,23].

Subsequently, we optimized the azolylacryloyl molecules to increase their pharmacologic activities while maintaining the Michael acceptor reactive center [4,25]. The first modification, yielding the MMA-100 series of compounds (with MMA-102 as an example, Figure 1), entailed incorporating an imidazolyl Michael acceptor reactive center and replacing the benzene ring of the predecessor KAUS compounds with a pyridine ring to enhance solubility and RBC partitioning. Increased RBC partitioning has previously been reported for aromatic aldehydes, where substituting a benzene ring with a pyridine ring significantly improved the partitioning of compounds into RBCs [4,25].

The second modification, yielding the MMA-200 series of compounds (with MMA-202 as an example, Figure 1), was aimed at increasing potency by replacing the imidazolyl Michael acceptor reactive group with a non-imidazolyl alternative to reduce potential steric hindrance at the βCys93 binding site [4,25]. For covalent interaction to form between the KAUS molecules and βCys93, the imidazole moiety of the Michael acceptor group has to move closer to the –SH group of βCys93 while avoiding steric contact with the surrounding binding site residues. As expected, the terminal enone derivatives in the MMA-200 series exhibited improved antisickling activity compared to the azolylacryloyl MMA-100 compounds [4,25], suggesting that Michael acceptor compounds with a non-imidazolyl reactive center are more effective. In this study, we synthesized another generation of Michael acceptor compounds by retaining the non-imidazolyl reactive center while introducing pyridine or furan substituents to enhance interactions with the binding pocket around βCys93 and/or improve RBC partitioning. In addition, increasing the size of the Michael acceptors may decrease their off-target binding. Four compounds, MMA-206 to -209, were synthesized based on the new design strategies (Figure 1 and Figure 2), as described in the experimental section and also briefly discussed below.

## 2. Results

### 2.1. Synthetic Strategy for MMA-206, MMA-208, and MMA-209

The pharmacophoric feature common to the title compounds MMA-206 to -209 is a 1-phenylpropenone (α,β-unsaturated ketone) system (see Figure 1), which was constructed by the addition of a carbon nucleophile (vinyl magnesium bromide) to a Weinreb amide at low temperature (−78 °C to 0 °C). Use of the Weinreb amide strategy was essential to selectively stop at the ketone stage. Following the initial addition, the resulting tetrahedral intermediate cannot undergo a second addition, as it collapses to release a neutral amide, yielding the desired ketone product [27].

A distinguishing feature of MMA-206, MMA-208, and MMA-209 is the presence of heterocyclic (pyridyl or furanyl) substituents connected via an ether linkage to the aromatic ring. These moieties were incorporated into the structures using either Mitsunobu coupling [28,29,30] (to introduce substituents in MMA-206 and MMA-208, Figure 1) or Williamson ether synthesis (for MMA-206’s final step in Figure 1 and to prepare intermediate 14 in Figure 2 for the synthesis of MMA-207).

The synthesis of MMA-206 started by converting carboxylic acid 1 into the Weinreb amide 3 using a direct amide coupling procedure. To complete the synthesis of MMA-206, the Weinreb amide was reacted with vinyl magnesium bromide to obtain enone 5, followed by a base-catalyzed Williamson reaction to yield MMA-206 in an overall yield of 15.2%. The base-catalyzed Williamson method was more practical due to the availability of ((6-bromomethyl)pyridin-2-yl)methanol (reagent 6).

The synthesis of MMA-207 (Figure 2) started with a different carboxylic acid (12, 2,5-dihydroxybenzoic acid). This acid was first protected by conversion to the methyl ester 13 (via reaction with methanol under acidic catalysis with SOCl_2_), thus avoiding methylation of the phenolic groups. The ester 13 was reacted with an equimolar amount of bromomethylpyridyl reagent 6 under Williamson conditions to obtain intermediate 14. This method avoided alkylation at the ortho-phenolic position, as alkylation occurred preferentially at the 5-hydroxy position. The regioselectivity of this reaction might be attributed to reduced availability of the 2-hydroxy group due to its engagement in an intramolecular hydrogen bond with the ortho ester group of 13 [31]. The protective methyl ester was removed by hydrolysis to obtain acid 15, and the synthesis was then completed via the Weinreb protocol as detailed above to obtain MMA-207 in 4.3% yield over five steps.

The synthesis of MMA-208 (Figure 1) required the preparation of a non-commercially available alcohol (intermediate 10). The synthesis of 10 was commenced from 5-HMF (7) by linking it with phenol (Mitsunobu reaction) [28,29,30], followed by NaBH_4_ reduction (see experimental section for details). Linking alcohol 10 to phenol 3 was achieved using a Mitsunobu reaction to obtain intermediate 11, which was then reacted with vinylmagnesium bromide to provide MMA-208 in 9.4% yield over three steps.

In a similar strategy, the synthesis of MMA-209 (Figure 1) started from the Weinreb amide 3 and yielded the final product in four steps, with an overall yield of 1.8%. In this synthesis, 5-HMF, a well-studied antisickling agent, was used to install a furfuryl group onto phenol 3.

### 2.2. MMA Compounds React with βCys93 of Hb

The MMA compounds are expected to bind to βCys93 of Hb to exert their pharmacologic effects, which were evaluated through a disulfide exchange reaction between 5,5′-dithiobis-(2-nitrobenzoic acid) (DTNB) and the sulfhydryl (–SH) group of βCys93. Although Hb contains six cysteine residues, only the two βCys93, one on each β subunit are known to be solvent-accessible and found to be reactive toward Michael acceptor compounds, thiols, or isothiocyanates [22,25]. The disulfide exchange reaction of the negative control experiment (Hb with an equivalent amount of DMSO) confirmed βCys93 residues to be the only two with accessible thiols (Figure 2). In the presence of MMA-206 or MMA-207, only about 10% of the βCys93 SH groups were observed to be freely available after 3 h incubation at room temperature, suggesting 90% of the βCys93 SH groups had reacted with the compounds (Figure 2). This compares with 70% reactivity of the positive control ECA. MMA-208 and MMA-209 showed 60% and 20% reactivity with the βCys93 SH groups, respectively. Since cell-free Hb was used in this experiment, it is evident that the pyridine substituent in MMA-206 and MMA-207 may have enhanced the reactivity when compared to the furan substituent in MMA-208 and MMA-209 and to the positive reference ECA. The methoxy and hydroxymethyl substituents on the phenyl rings of MMA-206 and MMA-207, respectively appear to confer similar reactivity to the compounds. A comparison of MMA-208 and MMA-209 suggests that the phenoxy substituent in MMA-208 enhances the compound’s reactivity more than the hydroxymethyl substituent in MMA-209.

### 2.3. MMA Compounds Increase Hb Oxygen Affinity with Sustained Effects

Aromatic aldehydes are known to suffer from oxidative metabolism by either aldehyde dehydrogenase or aldehyde oxidase in the liver, blood, and other tissues, leading to a short half-life and suboptimal bioavailability [4,32,33]. Our previous time-dependent OEC studies using whole blood have shown that Michael acceptor compounds provide much longer activities, presumably due to resistance to metabolic oxidation and the irreversible nature of their covalent interaction with Hb [22,23,25]. To determine whether the MMA compounds are also metabolically stable, we conducted time-dependent OEC studies with MMA-206 and MMA-209 using whole blood [25]. At defined time points (1.5, 3, 6, 8, 12, and 24 h), aliquot samples were drawn and subsequently analyzed for their P_50_-shifts relative to the initial P_50_ value using three-point tonometry. The results are presented in Figure 3. We observed sustained P_50_-shift activity throughout the 24 h experiment, with peak value at 6 h. Previous OEC studies using whole blood with 5-HMF, a reversible covalent aromatic aldehyde Hb binder, showed its effect to peak at 1.5 h and then steadily decline to base line at 24 h [34], suggesting significant RBC metabolism. Although other metabolic pathways, including hepatic metabolism, influence the pharmacologic activity of compounds, the whole blood experiment largely mimics the in vivo activity of aromatic aldehydes. Therefore, MMA compounds are expected to exhibit superior pharmacokinetic profiles in vivo.

### 2.4. MMA Compounds Increase HbS Oxygen Affinity and Inhibit RBC Sickling

In this study, we tested MMA-206 to -209, as well as the positive control MMA-202, in a dose-dependent manner (0.5 mM, 1.0 mM, and 2.0 mM) for their abilities to increase Hb oxygen affinity (left-shift P_50_) and inhibit deoxygenation-induced RBC sickling, as previously published [25]. The results, summarized in Table 1 and Figure 4, demonstrate a dose-dependent inhibition of sickling and an increase in hemoglobin oxygen affinity, especially at the 1 mM and 2 mM concentrations.

Based on *t*-test analysis, MMA-206 exhibited the most statistically significant sickling inhibition, with 52% and 65% inhibition at 1 mM and 2 mM, respectively (Table 1, Figure 4a). This compares to ~50% sickling inhibition observed with the positive control MMA-202 at 2 mM. In contrast, the other MMA compounds (MMA-209, MMA-208, and MMA-207) demonstrated lower sickling inhibition activity, ranging from ~35% to 43% at 2 mM and ~26 to 29% at 1 mM, with no statistically significant differences among their inhibition values. Prior to this study, MMA-202 was reported as the most potent antisickling Michael acceptor compound [25]. Our findings demonstrate a significant improvement in sickling inhibition potency with MMA-206 compared to other compounds in this class, including the previously reported most potent Michael acceptor compound, MMA-202. Nonetheless, MMA-202 still exhibits greater potency than MMA-207, MMA-208, and MMA-209.

Similar to the sickling inhibition study, the OEC study with the MMA compounds also demonstrated a dose-dependent effect on Hb oxygen affinity, particularly at the 1 and 2 mM concentrations (Table 1, Figure 4b). At 2 mM, MMA-206 increased Hb oxygen affinity by 28%, while MMA-207, MMA-208, and the positive control MMA-202 each produced an increase of approximately 20%, with MMA-209 showing a more modest increase of 12%. However, unlike the sickling inhibition results, where MMA-206 clearly demonstrated superior potency, *t*-test analysis of the OEC data revealed no statistically significant differences in P_50_ shift potency among all compounds at any concentration.

## 3. Discussion

In this study, we rationally designed and synthesized four Michael acceptor compounds, including MMA-206 to -209, and investigated their pharmacological activities and mechanisms of actions. Our long-term objective has been to improve the pharmacokinetic properties of antisickling aromatic aldehydes by replacing the reactive but metabolically unstable aldehyde group with a Michael acceptor group. The reversible nature of the Schiff-base interaction between the aromatic aldehyde and Hb results in the eventual irreversible loss of the dissociated aldehyde compound. We anticipated that modification of the aldehyde group to a Michael acceptor group would not only enhance the metabolic stability of the compounds but also enable irreversible covalent binding to hemoglobin, thereby prolonging the drug’s half-life and reducing both the required therapeutic dose and dosing frequency. Expectedly, the in vitro time-dependent hemoglobin oxygen equilibrium study, a reliable predictor of the metabolic stability of antisickling agents, demonstrated that the Michael acceptor compounds maintained their activity over an extended period, whereas the biological activity of the aromatic aldehyde comparator, 5-HMF, declined rapidly due to metabolic degradation of the aldehyde group. Consistent with these findings, 5-HMF failed in clinical studies, partly due to its poor oral bioavailability and short half-life of approximately one hour.

A second objective of the study was to enhance the potency of antisickling Michael acceptor compounds, which was achieved with MMA-206. The improved antisickling activity of MMA-206 may, in part, be attributed to its strong reactivity with βCys93 (Figure 2). Nonetheless, antisickling potency may not necessarily correlate with compound reactivity with βCys93. For example, the pyridinyl analog MMA-207, despite exhibiting comparable βCys93 reactivity to that of the structurally similar pyridinyl analog MMA-206, showed significantly weaker inhibition of sickling—comparable to the furan analogs (MMA-208 and MMA-209), which have lower βCys93 reactivity. Likewise, despite MMA-208 exhibiting significantly higher βCys93 reactivity than MMA-209, both furan analogs showed similar sickling inhibition potency. The key difference between MMA-208 and MMA-209 lies in their substitutions on the furan ring—phenoxy for MMA-208 and hydroxymethyl for MMA-209. Given its greater hydrophobicity, one would have expected MMA-208 to partition more efficiently into RBCs and exhibit stronger sickling inhibition compared to the less hydrophobic MMA-209. Antisickling activity is driven by two primary mechanisms: (1) increasing Hb oxygen affinity (left-shifting Hb oxygen equilibrium) by disrupting the T-state-stabilizing salt-bridge interaction between βAsp94 and βHis146, thereby increasing the concentration of non-polymer-forming, oxygenated HbS, and (2) direct destabilization of HbS polymers through compound binding to the protein surface. Higher reactivity may not necessarily enhance Hb oxygen affinity, unless it results in more effective disruption of the βAsp94–βHis146 interaction and/or greater polymer destabilization. This may account for the lack of a clear correlation between compound reactivity and antisickling activity. Additional factors contributing to the inconsistent correlation between reactivity and antisickling potency may include differences in RBC partitioning. For instance, MMA-207, which contains a hydroxymethyl substitution on the phenyl ring, is more polar and may partition less efficiently into RBCs to bind Hb. In contrast, MMA-206, with a less polar methoxy substitution, is expected to exhibit greater RBC partitioning and, consequently, more effective Hb binding. It is also important to note that the βCys93 reactivity studies were performed using cell-free hemoglobin, whereas the sickling assays were conducted in intact RBCs. Additionally, differences in the position of the pyridyl substituent may play a role; in MMA-206, the substituent is in the para position relative to the Michael acceptor, while in MMA-207, it is in the meta position, potentially affecting binding orientation and reactivity.

It is evident that all compounds increase Hb oxygen affinity, which partly explains their ability to prevent deoxygenation-induced RBC sickling. Nonetheless, there is an apparent discrepancy in the OEC (oxygen equilibrium curve) and sickling inhibition results, which warrants further discussion. First, despite the significant differences in antisickling potency between MMA-206 and the other MMA compounds, all the compounds appear to increase Hb oxygen affinity to a similar extent. For example, although MMA-206 exhibits nearly twofold higher antisickling activity compared to some of the other MMA compounds, it does not demonstrate a proportionally greater increase in Hb oxygen affinity.

Second, the relatively modest P_50_ shifts observed for MMA-206 to -209 (28%, 20%, 18%, and 12% at 2 mM, respectively) do not fully account for their substantial sickling inhibition activities (65%, 35%, 39%, and 43% inhibition at 2 mM, respectively). These findings contrast with previously studied antisickling agents such as 5-HMF and Voxelotor [4,13,14,15,16], where a strong correlation between antisickling potency and P_50_ shift was observed. This suggests that enhancing Hb oxygen affinity may not be the sole mechanism underlying the antisickling activity (i.e., the O_2_-dependent mechanism) of these compounds, and that an additional, O_2_-independent mechanism may be contributing.

Indeed, a similarly weak correlation between P_50_ shift and antisickling potency has been reported for other antisickling agents, such as the aromatic aldehyde VZHE-039 [4,11], one of the most potent RBC sickling inhibitors identified to date. The enhanced antisickling activity of VZHE-039 has been attributed to a secondary mechanism involving direct polymer destabilization that operates independently of oxygen (i.e., it remains effective even under anoxic conditions) [4,11]. VZHE-039 binds at the α-cleft of Hb, forming Schiff-base interactions with the αVal1 amines and inducing perturbations in the surface-exposed αF-helix, thereby disrupting HbS polymerization or interfering with HbS–HbS interactions.

We recently reported a similar observation of dual O_2_-dependent and O_2_-independent antisickling activities in another class of Michael acceptor compounds that bind to βCys93 [34]. Nonetheless, those compounds showed a significantly less potent antisickling effect than the current MMA compounds reported here.

## 4. Materials and Methods

### 4.1. Study Approval

At Virginia Commonwealth University (VCU), normal whole blood (AA) was collected from adult donors (>18 years) after informed consent, in accordance with regulations of the Institutional Review Board (IRB) at VCU for Protection of Human Subjects (IRB #HM1). At the Children’s Hospital of Philadelphia (CHOP), leftover blood samples from individuals with homozygous sickle cell (SS) who had not been recently transfused, were obtained and utilized based on an approved IRB protocol (IRB# 11-008151) by the IRB, with informed consent. All experimental protocols and methods were performed in accordance with institutional (VCU and CHOP) regulations.

### 4.2. Compound Synthesis

Unless otherwise noted, reagents and solvents were used as purchased from commercial suppliers. Solvent removal was accomplished usually using a rotary evaporator at ~15 mm Hg pressure unless otherwise specified. Thin-layer chromatography (TLC) was performed using silica-gel 60 plates F254 (Merck KGaA, Darmstadt, Germany) and visualized by UV light (254 nm). Column chromatography and flash were carried out using silica gel (60–203 mesh and 40–60 mesh, respectively), unless otherwise specified. Proton nuclear magnetic resonance (^1^H NMR) spectra were recorded on a Bruker Avance II 300 MHz spectrometer (Bruker BioSpin, Billerica, MA, USA) at ambient temperature. Chemical shifts were calibrated to residual solvent peaks or internal tetramethylsilane (TMS). All target compounds and key intermediates were confirmed by ^1^H NMR (and ^13^C NMR where applicable) in appropriate deuterated solvents (e.g., DMSO-d_6 or CDCl_3), verifying their structures. Sample purity was assessed by ultra-performance liquid chromatography–mass spectrometry (UPLC–MS) using a Waters Acquity H-Class system equipped with a single-quadrupole QDa mass detector (Waters Corporation, Milford, MA, USA). UPLC separations employed a Phenomenex Kinetex EVO C_18 column (50 × 3 mm, 2.6 µm) (Phenomenex Inc, Torrance, CA, USA) with UV detection at 220 nm (and 254 nm). A short linear gradient (2–5 min runtime) of water and acetonitrile was used, under either acidic conditions (0.1% trifluoroacetic acid) or neutral buffered conditions (20 mM ammonium formate, pH 7.4), selected according to compound polarity. The UPLC–UV chromatograms typically showed a single major peak for each compound, and the QDa MS spectra displayed the expected molecular ion ([M+H]^+^ or Na^+^ adduct) for the product, confirming each compound’s identity and homogeneity. All final Michael acceptor analogues (MMA-206 through MMA-209) showed high purity by UPLC (generally >95% area under the curve) and gave NMR spectra consistent with their proposed structures.


*Preparation of 4-Hydroxy-N,3-dimethoxy-N-methylbenzamide (3)*


Vanillic acid (1, 5.0 g, 29.7 mmol, 1.0 equiv) was dissolved in dry DMF (37.5 mL), and N,O-Dimethylhydroxylamine HCl (2, 3.57 g, 36.6 mmol, 1.23 equiv) was added. The mixture was cooled to 0 °C, and HOBt (602 mg, 4.46 mmol, 0.15 equiv), N-Methylmorpholine (NMM, 6.86 mL, 62.4 mmol, 2.1 equiv), and 1-Ethyl-3-(3-dimethylaminopropyl)carbodiimide (EDCI, 6.84 g, 35.7 mmol, 1.2 equiv) were added. The resultant mixture was stirred at 0 °C for 75 min and at rt overnight. The progress of the reaction was monitored by TLC (dichloromethane-MeOH = 95:5) and LCMS analyses. After completion, the mixture was diluted with dichloromethane and extracted with citric acid (10%, twice). The organic phase was dried over Na_2_SO_4_, filtered and concentrated. The crude product was purified by column chromatography (eluent: 0–5% MeOH in dichloromethane) and recrystallized from Diisopropyl ether (DIPE) to afford the intermediate compound 4-Hydroxy-N,3-dimethoxy-N-methylbenzamide (3, 4.71 g, 75% yield) as a white crystal. ^1^H-NMR (300 MHz, DMSO-d_6_) δ 9.57 (s, 1H), 7.20 (d, J = 1.7 Hz, 1H), 7.15 (dd, J = 8.2, 1.8 Hz, 1H), 6.81 (d, J = 8.2 Hz, 1H), 3.79 (s, 3H), 3.57 (s, 3H), 3.22 (s, 3H). MS *m*/*z* 212 [M+H]^+^; UPLC-MS using (220 nm) 84% (AUC). Melting point: 101–102 °C.


*Synthesis of MMA-206*



*Preparation of 1-(4-Hydroxy-3-methoxyphenyl)prop-2-en-1-one (5)*


The Weinreb amide 3 (1.5 g, 7.1 mmol, 1.0 equiv) was dissolved in anhydrous THF (35 mL) under argon atmosphere, and the resulting solution was cooled to −78 °C. Then, a solution of vinyl magnesium bromide (4, 28.4 mL, 28.4 mmol, 4.0 equiv, 1M in THF) was added dropwise. After complete addition, the cooling bath was removed, and the mixture was allowed to warm to room temperature. The progress of the reaction was monitored by TLC (dichloromethane-MeOH = 95:5) and LCMS analyses. After completion, the mixture was poured into a vigorously stirred solution of 1M HCl (100 mL) and stirred for 15 min. The resulting mixture was extracted with dichloromethane twice, and the combined organic layer was dried over MgSO_4_, filtered and concentrated. The crude product was purified by column chromatography (eluent: 0–20% EtOAc in n-heptane) to afford the intermediate compound 1-(4-Hydroxy-3-methoxyphenyl)prop-2-en-1-one (5, 0.98 g, 78% yield) as a light brown oil. ^1^H-NMR (300 MHz, DMSO-d_6_) δ 10.11 (s, 1H), 7.60 (d, J = 8.4 Hz, 1H), 7.52 (s, 1H), 7.43 (dd, J = 16.9, 10.4 Hz, 1H), 6.89 (d, J = 8.3 Hz, 1H), 6.29 (d, J = 16.9 Hz, 1H), 5.88 (dd, J = 10.3, 1.7 Hz, 1H), 3.84 (s, 3H). MS *m*/*z* 179 [M+H]^+^; UPLC-MS (220 nm) 99% (AUC).


*Preparation of 1-(4-((6-(Hydroxymethyl)pyridin-2-yl)methoxy)-3-methoxyphenyl)prop-2-en-1-one (MMA-206)*


The enone intermediate 5 (0.3 g, 1.68 mmol, 1.0 equiv) was combined with (6-(bromomethyl)pyridin-2-yl)methanol (6) (0.544 g, 2.69 mmol, 1.6 equiv) and K_2_CO_3_ (0.7 g, 5.05 mmol, 3.0 equiv) in MeCN (5 mL) and stirred at 60 °C for 5 h. The progress of the reaction was monitored by TLC (cyclohexane-EtOAc = 1:2) and LCMS analyses. After completion, the mixture was filtered off, and the inorganic salt was washed with dichloromethane. The filtrate was concentrated under reduced pressure, and the crude product was purified by column chromatography (eluent: 0–50% EtOAc in cyclohexane) to afford the desired product (MMA-206, 130 mg, 26% yield) as an off-white crystal. ^1^H-NMR (300 MHz, CDCl_3_) δ 7.73 (t, J = 7.7 Hz, 1H), 7.62 (d, J = 1.6 Hz, 1H), 7.54 (dd, J = 8.3, 1.7 Hz, 1H), 7.45 (d, J = 7.5 Hz, 1H), 7.25–7.07 (m, 2H), 6.92 (d, J = 8.4 Hz, 1H), 6.44 (dd, J = 17.0, 1.6 Hz, 1H), 5.89 (dd, J = 10.5, 1.5 Hz, 1H), 5.38 (s, 2H), 4.80 (s, 2H), 4.00 (s, 3H), 3.72 (s, 1H). 13C-NMR (300 MHz, CDCl_3_) δ 189.18, 158.64, 155.52, 152.13, 149.69, 137.68, 131.93, 131.01, 129.42, 123.24, 119.83, 119.57, 112.09, 111.32, 71.25, 63.97, 56.15. MS *m*/*z* 300 [M+H]^+^; UPLC-MS (220 nm) 95% (AUC). Melting point: 81–82 °C.


*Synthesis of MMA-207*



*Preparation of Methyl 2,5-dihydroxybenzoate (13)*


2,5-Dihydroxybenzoic acid (12, 2.0 g, 12.98 mmol, 1.0 equiv) was dissolved in MeOH (30 mL), SOCl2 (1.42 mL, 19.5 mmol, 1.5 equiv) was added dropwise at 0 °C, and the reaction mixture was refluxed overnight. The progress of the reaction was monitored by TLC (dichloromethane-MeOH = 9:1 + NH_4_OH) and LCMS analyses. After completion, the mixture was evaporated, and the crude product was co-evaporated with toluene to give the intermediate compound Methyl 2,5-dihydroxybenzoate (13, 2.18 g, 99.9% yield) as a white crystal, which was used in the next step without further purification. ^1^H-NMR (300 MHz, DMSO-d_6_) δ 9.92 (s, 1H), 9.21 (s, 1H), 7.14 (d, J = 3.0 Hz, 1H), 6.98 (dd, J = 8.9, 3.0 Hz, 1H), 6.82 (d, J = 8.8 Hz, 1H), 3.87 (s, 3H). MS *m*/*z* not detected (220 nm) 99% (AUC). Melting point: 92–93 °C.


*Preparation of Methyl 2-hydroxy-5-((6-(hydroxymethyl)pyridin-2-yl)methoxy)benzoate (14)*


Starting compound (13, 1.69 g, mmol, 1.0 equiv) was combined with (6-(bromomethyl)pyridin-2-yl)methanol (6, 2.03 g, 10.0 mmol, 1.0 equiv) and K_2_CO_3_ (2.5 g, 18.1 mmol, 1.8 equiv) in dry acetone (40 mL) and stirred at 50 °C for 48 h. The progress of the reaction was monitored by TLC (dichloromethane:MeOH-95:5) and LCMS analyses. After completion, the mixture was filtered off, and the inorganic salt was washed with dichloromethane. The filtrate was evaporated under reduced pressure, and the concentrate was dissolved in dichloromethane, washed with water (twice), dried over anhydrous MgSO_4_, and evaporated. The crude product was purified by column chromatography (eluent: 0–5% MeOH in dichloromethane) to afford the intermediate compound Methyl 2-hydroxy-5-((6-(hydroxymethyl)pyridin-2-yl)methoxy)benzoate (14, 1.32 g, 45% yield) as a yellow solid. ^1^H-NMR (300 MHz, DMSO-d_6_) δ 10.09 (s, 1H), 7.83 (t, J = 7.7 Hz, 1H), 7.53–7.32 (m, 3H), 7.26 (dd, J = 9.0, 3.1 Hz, 1H), 6.94 (d, J = 9.0 Hz, 1H), 5.44 (t, J = 5.8 Hz, 1H), 5.10 (s, 2H), 4.57 (d, J = 5.5 Hz, 2H), 3.89 (s, 3H). MS *m*/*z* 290 [M+H]^+^; UPLC-MS (220 nm) 75% (AUC). Melting point: 85–86 °C.


*Preparation of 2-Hydroxy-5-((6-(hydroxymethyl)pyridin-2-yl)methoxy)benzoic acid (15)*


Starting ester (14, 1.27 g, 4.4 mmol, 1.0 equiv) was dissolved in MeOH (14 mL), 1M LiOH solution (15.5 mL, 15.4 mmol, 3.5 equiv) was added, and the mixture was stirred at 45 °C for 6 h. The progress of the reaction was monitored by TLC (dichloromethane-MeOH = 95:5 and 95:5 + AcOH) and LCMS analyses. After completion, the mixture was poured into water, its pH was adjusted to 2–3 by the addition of 1M aq. HCl, and the precipitate was filtered off, washed with water, and dried to obtain the intermediate compound 2-Hydroxy-5-((6-(hydroxymethyl)pyridin-2-yl)methoxy)benzoic acid (15, 1.07 g, 88% yield) as a white crystal. ^1^H-NMR (300 MHz, DMSO-d6) δ 10.99 (s, 1H), 7.83 (t, J = 7.7 Hz, 1H), 7.49–7.29 (m, 3H), 7.24 (dd, J = 9.0, 3.1 Hz, 1H), 6.91 (d, J = 9.0 Hz, 1H), 5.42 (s, 1H), 5.10 (s, 2H), 4.57 (s, 2H). MS *m*/*z* 276 [M+H]^+^; UPLC-MS (220 nm) 95% (AUC). Melting point: 242–243 °C.


*Preparation of 2-Hydroxy-5-((6-(hydroxymethyl)pyridin-2-yl)methoxy)-N-methoxy-N-methylbenzamide (16)*


This Weinreb amide was prepared according to the procedure described for the preparation of 3. The crude product was purified by column chromatography (eluent: 0–5% MeOH in dichloromethane) to afford the intermediate compound 2-Hydroxy-5-((6-(hydroxymethyl)pyridin-2-yl)methoxy)-N-methoxy-N-methylbenzamide (16, 0.81 g, 66% yield) as a white crystal. MS *m*/*z* 319 [M+H]^+^; UPLC-MS (220 nm) 95% (AUC), which was used for the next reaction without further purification.


*Preparation of 1-(2-Hydroxy-5-((6-(hydroxymethyl)pyridin-2-yl)methoxy)phenyl)prop-2-en-1-one (MMA-207)*


The intermediate 16 (0.81 g, mmol, 1.0 equiv) was dissolved in anhydrous THF (12.5 mL) under argon atmosphere, and the resulting solution was cooled to −78 °C. Then, a solution of vinyl magnesium bromide (4, 10.2 mL, 10.2 mmol, 4.0 equiv, 1M in THF) was added dropwise. After complete addition, the mixture was stirred at −78 °C for 2.5 h; then, it was allowed to warm to 0 °C while stirring. The progress of the reaction was monitored by LCMS analysis. After completion, the mixture was poured into a vigorously stirred cold solution of 1 M HCl (50 mL) and stirred for 15 min. The resulting mixture was extracted with dichloromethane twice and the combined organic layers dried over MgSO_4_ and concentrated. The crude product was purified by preparative HPLC to afford the final compound 1-(2-Hydroxy-5-((6-(hydroxymethyl)pyridin-2-yl)methoxy)phenyl)prop-2-en-1-one (MMA-207, 0.12 g, 16.5% yield) as a yellowish crystal. ^1^H-NMR (300 MHz, CDCl_3_) δ 12.13 (s, 1H), 7.76 (t, J = 7.7 Hz, 1H), 7.46 (d, J = 7.6 Hz, 1H), 7.38 (d, J = 3.0 Hz, 1H), 7.32–7.16 (m, 3H), 6.99 (d, J = 9.1 Hz, 1H), 6.56 (dd, J = 16.9, 1.6 Hz, 1H), 5.99 (dd, J = 10.6, 1.5 Hz, 1H), 5.21 (s, 2H), 4.81 (s, 2H), 3.65 (s, 1H). 13C-NMR (300 MHz, CDCl_3_) δ 194.08, 170.08, 158.62, 158.34, 155.99, 150.47, 137.64, 131.02, 130.67, 125.34, 120.08, 119.58, 118.96, 114.19, 71.48, 64.02. MS *m*/*z* 286 [M+Na]^+^; UPLC-MS using (220 nm) 95% (AUC). Melting point: 75–78 °C.


*Synthesis of MMA-208*



*Preparation of 5-(Phenoxymethyl)furan-2-carbaldehyde (10)*


Starting aldehyde 5-hydroxymethylfurfural (5-HMF, 7, 2.0 g, 15.86 mmol, 1.0 equiv) was dissolved in THF (104 mL); then, phenol (2.24 g, 1.5 equiv), TPP (2.24 g, 23.8 mmol, 1.5 equiv), and DIAD (4.67 mL, 23.8 mmol, 1.5 equiv) were added. The resultant mixture was stirred at rt overnight. After completion, the solvent was evaporated under reduced pressure, and the residue was partitioned between dichloromethane and 10% aq. NaOH solution. After separation, the organic phase was dried over MgSO_4_, filtered, and evaporated. The crude product was purified by column chromatography (eluent: 0–30% EtOAc in cyclohexane) to afford the intermediate product 5-(Phenoxymethyl)furan-2-carbaldehyde (10, PMFC). 1H-NMR (CDCl_3_, 300 MHz): δ 9.60 (s, 1H), 7.54 (d, J = 3.5 Hz, 1H), 7.32 (td, J = 8.0, 1.9 Hz, 2H), 7.04 (d, J = 7.9 Hz, 2H), 6.98 (t, J = 7.3 Hz, 1H), 6.87 (d, J = 3.5 Hz, 1H), 5.20 (s, 1H). MS *m*/*z* 203 [M+H]^+^; UPLC-MS (220 nm) 98% (AUC). Mp: 95–96 °C.

PMFC (1.69 g, 8.36 mmol, 1.0 equiv) was dissolved in a mixture of MeOH (15 mL) and dichloromethane (15 mL) under N_2_ atmosphere and cooled to 0 °C. NaBH_4_ (0.38 g, 10.0 mmol, 1.2 equiv) was added portion-wise, and the mixture was stirred at 0 °C for 15 min. Then, the cooling bath was removed, and the mixture was allowed to warm to room temperature and stirred for 1 h. The progress of the reaction was monitored by TLC (n-heptane-EtOAc = 3:2) and LCMS analyses. After completion, the reaction was quenched with water. The volatile solvent was removed under reduced pressure, and the residual aqueous solution was extracted with EtOAc. The organic layer was dried over MgSO_4_, filtered, and evaporated. The crude product was purified by column chromatography (eluent: 0–5% MeOH in dichloromethane) to afford the intermediate product 5-(Phenoxymethyl)furan-2-carbaldehyde (10, 1.56 g, 91% yield) as a yellowish solid. ^1^H-NMR (300 MHz, DMSO-d_6_) δ 7.30 (t, J = 8.0 Hz, 2H), 7.02 (d, = 8.0 Hz, 2H), 6.95 (t, J = 7.3 Hz, 1H), 6.50 (d, J = 3.1 Hz, 1H), 6.27 (d, J = 3.0 Hz, 1H), 5.24 (t, J = 5.3 Hz, 1H), 5.01 (s, 2H), 4.38 (d, J = 5.1 Hz, 2H). MS *m*/*z* 187 [M+H-H_2_O]+; UPLC-MS (220 nm) 97% (AUC). Melting point: 37–39 °C.


*Preparation of N,3-Dimethoxy-N-methyl-4-((5-(phenoxymethyl)furan-2-yl)methoxy)benzamide (11)*


The phenolic intermediate 3 (1.0 g, 4.74 mmol, 1.0 equiv) was dissolved in THF (30 mL); then, the alcohol reagent 10 (1.16 g, 5.68 mmol, 1.2 equiv), Triphenylphosphine (TPP, 1.86 g, 7.1 mmol, 1.5 equiv), and Diisopropyl azodicarboxylate (DEAD, 1.11 mL, 7.1 mmol, 1.5 equiv) were added. The resultant mixture was stirred at rt overnight. The progress of the reaction was monitored by TLC (dichloromethane-MeOH = 95:5) and LCMS analyses. After completion, the solvent was evaporated under reduced pressure, and the residue was partitioned between EtOAc and water. After separation, the organic phase was dried over MgSO_4_, filtered, and evaporated. The crude product was purified by column chromatography (eluent: 0–50% EtOAc in cyclohexane) to afford the intermediate compound N,3-Dimethoxy-N-methyl-4-((5-(phenoxymethyl)furan-2-yl)methoxy)benzamide (11, 1.81 g impure, 96% yield, contaminated with some triphenylphosphine oxide) as a yellowish oil. ^1^H NMR (300 MHz, DMSO-d_6_) δ 7.29 (dd, J = 16.2, 8.5 Hz, 3H), 7.22 (s, 1H), 7.16 (d, J = 8.2 Hz, 1H), 7.03 (d, J = 7.9 Hz, 2H), 6.96 (t, J = 7.2 Hz, 1H), 6.61 (s, 2H), 5.08 (d, J = 11.7 Hz, 4H), 3.78 (s, 3H), 3.57 (s, 3H), 3.23 (s, 3H). MS *m*/*z* 398 [M+H]^+^; (220 nm) 55% (AUC).


*Preparation of 1-(3-Methoxy-4-((5-(phenoxymethyl)furan-2-yl)methoxy)phenyl)prop-2-en-1-one (MMA-208)*


This final compound was prepared following the procedure described for the preparation of 5. Starting Weinreb amide 11 (1.0 g, 2.51 mmol, 1.0 equiv) was dissolved in anhydrous THF (12 mL) under argon atmosphere, and the resulting solution was cooled to −78 °C. Then, a solution of vinyl magnesium bromide (1M in THF, 10 mL, 10.1 mmol, 4.0 equiv) was added dropwise. After complete addition, the mixture was stirred at −78 °C for 2.5 h; then, it was allowed to warm to 0 °C and stirred for 30 min. The progress of the reaction was monitored by LCMS analysis. After completion, the mixture was poured into a vigorously stirred cold solution of 1 M HCl (50 mL) and stirred for 15 min. The resulting mixture was extracted with dichloromethane twice, and the combined organic layer was dried over MgSO_4_, filtered and concentrated. The crude product was purified by preparative HPLC to afford the final compound 1-(3-Methoxy-4-((5-(phenoxymethyl)furan-2-yl)methoxy)phenyl)prop-2-en-1-one (MMA-208, 120 mg, 13% yield) as a yellowish crystal. ^1^H-NMR (300 MHz, DMSO-d_6_) δ 7.71 (dd, J = 8.5, 1.7 Hz, 1H), 7.62–7.38 (m, 2H), 7.29 (dd, J = 14.3, 8.4 Hz, 3H), 7.03 (d, J = 7.9 Hz, 2H), 6.96 (t, J = 7.2 Hz, 1H), 6.62 (dd, J = 9.7, 3.1 Hz, 2H), 6.32 (dd, J = 16.9, 2.0 Hz, 1H), 5.92 (dd, J = 10.4, 1.9 Hz, 1H), 5.17 (s, 2H), 5.06 (s, 2H), 3.83 (s, 3H). 13C-NMR (300 MHz, DMSO-d_6_) δ 200.35, 164.78, 158.38, 152.32, 151.51, 150.56, 149.53, 132.63, 130.57, 130.01, 129.75, 123.66, 121.47, 115.21, 112.91, 112.57, 111.91, 111.40, 63.33, 62.75, 62.04, 56.00. MS *m*/*z* 387 [M+H+Na]^+^; UPLC-MS (220 nm) 98% (AUC). Melting point: 87–89 °C.


*Synthesis of MMA-209*



*Preparation of 4-((5-Formylfuran-2-yl)methoxy)-N,3-dimethoxy-N-methylbenzamide (8)*


The phenolic intermediate 3 (1.37 g, 6.5 mmol, 1.0 equiv) was dissolved in THF (41 mL); then, 5-(Hydroxymethyl)furfural (5-HMF, 7, 982 mg, 7.8 mmol, 1.2 equiv), TPP (2.55 g, 9.72 mmol, 1.5 equiv), and DIAD (1.91 mL, 9.72 mmol, 1.5 equiv) were added. The resultant mixture was stirred at rt overnight. The progress of the reaction was monitored by TLC (dichloromethane-MeOH = 95:5) and LCMS analyses. After completion, the solvent was evaporated under reduced pressure, and the residue was partitioned between EtOAc and water. After separation, the organic phase was dried over MgSO_4_, filtered, and evaporated. The crude product was purified by column chromatography (eluent: 0–60% EtOAc in cyclohexane) to afford the intermediate product 4-((5-Formylfuran-2-yl)methoxy)-N,3-dimethoxy-N-methylbenzamide (8, 1.39 g, 67% yield) as a yellowish-white solid. ^1^H-NMR (300 MHz, DMSO-d_6_) δ 9.62 (s, 1H), 7.55 (d, J = 3.5 Hz, 1H), 7.30–7.09 (m, 3H), 6.89 (d, J = 3.5 Hz, 1H), 5.24 (s, 2H), 3.79 (s, 3H), 3.57 (s, 3H), 3.24 (s, 3H). MS *m*/*z* 320 [M+H]^+^; UPLC-MS (220 nm) 96% (AUC). Melting point: 102–103 °C.


*Preparation of 4-((5-(Hydroxymethyl)furan-2-yl)methoxy)-N,3-dimethoxy-N-methylbenzamide (9)*


Starting aldehyde (8 1.365 g, 4.27 mmol, 1.0 equiv) was dissolved in MeOH (13.5 mL) under N_2_ atmosphere and cooled to 0 °C; then, NaBH4 (0.324 g, 8.55 mmol, 2.0 equiv) was added portion-wise, and it was stirred at 0 °C for 15 min. Then, the cooling bath was removed, and the mixture was allowed to warm to room temperature and stirred for 1 h. The progress of the reaction was monitored by TLC (dichloromethane-MeOH = 95:5) and LCMS analyses. After completion, the reaction was quenched with water. The volatile solvent was removed under reduced pressure, and the residual aqueous solution was extracted with EtOAc. The organic layer was dried over MgSO_4_, filtered, and evaporated. The crude product was purified by column chromatography (eluent: 0–5% MeOH in dichloromethane) to afford the intermediate product 4-((5-(Hydroxymethyl)furan-2-yl)methoxy)-N,3-dimethoxy-N-methylbenzamide (9, 763 mg, 55% yield) as an off-white solid. ^1^H-NMR (300 MHz, DMSO-d_6_) δ 7.29–7.09 (m, 3H), 6.53 (d, J = 3.1 Hz, 1H), 6.29 (d, J = 3.0 Hz, 1H), 5.27 (t, J = 5.8 Hz, 1H), 5.05 (s, 2H), 4.39 (d, J = 5.7 Hz, 2H), 3.77 (s, 3H), 3.58 (s, 3H), 3.24 (s, 3H). MS *m*/*z* 344 [M+Na]^+^; UPLC-MS using Waters method 5min_slow_A1B1_EVO (220 nm) 95% (AUC). Melting point: 104–105 °C.


*Preparation of 1-(4-((5-(Hydroxymethyl)furan-2-yl)methoxy)-3-methoxyphenyl)prop-2-en-1-one (MMA-209)*


Starting compound (16, 730 mg, 2.27 mmol, 1.0 equiv) was dissolved in anhydrous THF (11 mL) under argon atmosphere, and the resulting solution was cooled to −78 °C. Then, a solution of vinyl magnesium bromide (4, 9.1 mL, 9.1 mmol, 4.0 equiv, 1M in THF) was added dropwise. After complete addition, the mixture was stirred at −78 °C for 2.5 h; then, it was allowed to warm to 0 °C and stirred for 30 min. The progress of the reaction was monitored by LCMS analysis. After completion, the mixture was poured into a vigorously stirred cold solution of 1 M HCl (40 mL) and stirred for 15 min. The resulting mixture was extracted with dichloromethane twice, and the combined organic layer was dried over MgSO_4_, and concentrated. The crude product was purified by preparative HPLC and lyophilized to afford the final compound 1-(4-((5-(Hydroxymethyl)furan-2-yl)methoxy)-3-methoxyphenyl)prop-2-en-1-one (MMA-209, 32 mg, 5% yield) as a white crystal. ^1^H-NMR (300 MHz, DMSO-d_6_) δ 7.77–7.05 (m, 4H), 6.54 (d, J = 3.0 Hz, 1H), 6.29 (d, J = 2.8 Hz, 1H), 5.33–5.17 (m, 1H), 5.15–4.96 (m, 2H), 4.39 (d, J = 4.6 Hz, 2H), 3.88–3.70 (m, 3H). MS *m*/*z* 311 [M+Na]^+^; UPLC-MS (220 nm) 85% (AUC). Melting point: 89–91 °C.


*Reactivity of MMA Compounds with Accessible Sulfhydryl Groups in Hb*


The accessible sulfhydryl (SH) groups in Hb and their reactivity with the MMA compounds was quantified using a disulfide exchange reaction between the SH group of βCys93 and 5,5′-dithiobis-(2-nitrobenzoic acid) (DTNB, Ellman’s reagent) by measuring absorbance at 412 nm (ε = 14,150 M^−1^ cm^−1^) [25]. An aqueous solution of Hb (50 μM in PBS) was mixed with MMA-206, MMA-207, MMA-208, or MMA-209, and the positive control ethacrynic acid (ECA) at 2mM final concentration in a final volume of 500 μL. The tubes were incubated at room temperature for 3h with shaking at 100rpm. The reaction mixture was then transferred to a microfiltration centrifugal tube (MWCO 10 kDa) and centrifuged at 7000 rpm for 30 min at 4 °C to separate Hb from excess reagents. Modified Hb was further washed with PBS and centrifuged again to a reaction volume of 100 μL. A total of 25 μL of each Hb solution was added to 475 µL of 100 mM potassium phosphate buffer, pH 8.0, and incubated at 25 °C for 1 h (non-DTNB control samples). Another 25 μL of each Hb solution was added to the phosphate buffer (465 μL) with 10 μL of DTNB (10 mM) and incubated at 25 °C for 1 h. Before centrifuging the non-DTNB control samples, the absorbance of each sample was noted at 576 nm to determine the concentration of Hb in each sample. Both sets of tubes were centrifuged using different centrifugal filters (7000 rpm, 20 min, 4 °C) to collect the yellow filtrate (2-nitro-5-thiobenzoate), which was quantified by measuring an absorbance at 412 nm.


*In vitro Time-Dependent Hb Oxygen Equilibrium Curve Studies Using Normal Whole Blood*


As previously reported [25], normal whole blood samples (hematocrit 30%) in the presence of 2 mM concentration of the test compounds, MMA-206 and MMA-208, were incubated at 37 °C for 24 h with shaking (at 100 rpm). At 1.5, 3, 6, 12, and 24 h time intervals, aliquots of the mixture were removed and further incubated in TM8000 Thin film tonometer (Meon Medical Solutions, Graz, Austria) to equilibrate at oxygen tensions 6, 20, and 40 mmHg for approximately 10 min at 37 °C. The samples were then aspirated into an ABL 800 Automated Blood Gas Analyzer (Radiometer, Copenhagen, Denmark) to determine the partial pressure of oxygen (pO_2_) and Hb oxygen saturation values (SO_2_). The measured values of pO_2_ (mmHg) and SO_2_ at each oxygen tension values were then subjected to a non-linear regression analysis using the program Scientist (Micromath, Salt Lake City, UT, USA) to estimate P_50_, as previously reported [25]. The observed P_50_ shifts values in %P_50_ shifts were plotted as function of time (h).


*In vitro Antisickling and Oxygen Equilibrium Curve Studies Using Human Homozygous Sickle Cell (SS) Blood*


The compounds MMA-206 to -209 and the positive control MMA-202 were studied for their abilities to increase Hb affinity for oxygen and inhibit deoxygenated-induced RBC sickling (RBC morphology study), as previously published [25]. Briefly, homozygous SS blood (hematocrit 20%) suspensions were incubated under air in the absence or presence of 0.5 mM, 1.0 mM, and 2 mM concentration of test compounds at 37 °C for 1 h. Following, the suspensions were incubated under hypoxic condition (2.5% O_2_ gas/97.5% N_2_ gas) at 37 °C for 2 h. Aliquot samples were fixed with 2% glutaraldehyde solution without exposure to air and then subjected to microscopic morphological analysis. The residual samples were washed in phosphate-buffered saline and hemolyzed in hypotonic lysis buffer for the Hb oxygen equilibrium curve (OEC) experiment. For the OEC study, approximately 100 μL aliquot samples from the lysate were added to 4 mL of 0.1 M potassium phosphate buffer, pH 7.0, in a cuvette and subjected to hemoximetry analysis, using Hemox™ Analyzer (TCS Scientific Corp., New Hope, PA, USA) to assess P_50_ shifts.

## 5. Conclusions

Targeting Hb with aromatic aldehydes to prevent deoxygenation-induced HbS polymerization and RBC sickling has long been considered the gold standard for targeting Hb for SCD therapeutics [4,15]. While this approach has proven effective, as exemplified by the approval of Voxelotor [4,15], it also has limitations, particularly due to the poor PK properties associated with the metabolic instability of the aldehyde functional group. To address this challenge, our group has been developing alternative classes of compounds, including those featuring a Michael acceptor reactive center [4,22,23,25], designed to overcome the metabolic liabilities of aromatic aldehydes. This effort has culminated in the development of MMA-206, the most promising Michael acceptor compound identified to date. Not only is MMA-206 metabolically stable, but it also exhibits significantly greater antisickling potency, even more than the previously studied aromatic aldehyde, 5-HMF, which advanced to phase I/II clinical trials that terminated due in part to poor PK properties [4,35].

Importantly, MMA-206 appears to possess a secondary mechanism of antisickling activity that is independent of oxygen. We propose that this activity arises from the ability of the compound to disrupt HbS-HbS molecular interactions through its surface binding to the protein, thereby directly inhibiting HbS polymerization. Such direct polymer destabilizers are considered more effective therapeutics for SCD treatment, as their activity is not contingent upon oxygen availability [4]. In contrast, agents that act solely by increasing Hb oxygen affinity may be less effective in severely hypoxic conditions and become ineffective in treating vaso-occlusive crises [4]. Additionally, there are inherent limitations to elevating Hb oxygen affinity, as excessive shifts can hinder oxygen release to peripheral tissues, potentially impairing overall oxygen delivery. Further studies are planned to elucidate the multiple mechanisms of action of MMA-206, as well as to evaluate its in vivo efficacy and toxicity profile.

## Data Availability

The original contributions presented in this study are included in the article/Appendix A. Further inquiries can be directed to the corresponding author.

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
