# Peer review of "Michael Acceptor Compounds as Hemoglobin Oxygen Affinity Modulators for Reversing Sickling of Red Blood Cells"

_pharmaceuticals, 2025, doi:10.3390/ph18060783_

Round 1
Reviewer 1 Report
Comments and Suggestions for Authors
This is a well structured and mechanically rich study that explores the design and biological evaluation of Michael addition compounds targeting HbS polymerization and red blood cell sickling.
However, despite the promising results and insights there are few areas to be improved.
- The study uses BCys93 reactivity as a rational design, but the actual anti-sickling activity does not correlate consistently with the reactivity. The discrepancies suggest that the reactivity alone may not be reliable predictor for anti-sickling efficacy. More clear interpretation would strengthen the overall mechanistic insights and clarify the structure activity relationship.
- The claim of intramolecular hydrogen bonding and conformational change in MMA-207 is based on the inference (Fig.5) not supported by NMR or Molecular modeling. Without such data, the argument remains hypothetical and weakens the SAR discussion. The authors should either provide supporting data or clearly state the proposed intramolecular interactions are speculative.
- Authors should highlight the limitations of previously reported anti-sickling agents in terms of their poor ADME properties. Given the rational design of the new MMA compounds, it would strengthen the manuscript to include a comparative discussion on how these new analogs address earlier shortcomings. Specifically, the authors are encouraged to elaborate on any improvements in solubility, metabolic stability, or permeability and to comment on the drug likeliness of the new compounds. This comparison would provide a more comprehensive perspective on the new MMA series of compounds.
Reviewer 2 Report
Comments and Suggestions for Authors
The manuscript entitled “Michael Addition Compounds as Hemoglobin Oxygen Affinity Modulators for Reversing Sickling of Red Blood Cells” presents a well-structured and scientifically sound investigation, with a clear rationale and a focused objective. The authors successfully demonstrate that the synthesized compounds covalently modify hemoglobin at βCys93, prevent HbS polymerization, and exhibit sustained pharmacologic activity. Particularly, MMA-206 shows promising antisickling efficacy. This work thoughtfully extends the authors’ previous research on this class of compounds, offering valuable new insights.
However, some clarifications and refinements are needed to improve clarity, precision, and accessibility for a broader scientific audience.
Major comments:
- The introduction is currently too brief and would benefit from additional detail, particularly regarding the rationale behind the design of the new derivatives. I think it would be best to incorporate the rational and design strategy presented in Section 2.1 into the introduction, as it provides crucial context that strengthens the scientific foundation of the study.
-From a pharmacological perspective, it would be valuable to clarify why an irreversible covalent bond formed between a Michael acceptor and a cysteine residue (e.g., βCys93 of hemoglobin) is considered more advantageous than the reversible Schiff-base interaction between an aldehyde and an amine residue (e.g., αVal1), beyond the commonly cited issue of metabolic instability of aldehydes. A brief discussion addressing this comparison, or a justification for its exclusion, would enhance the manuscript's depth and provide important context for readers.
-Throughout the manuscript, the term “Michael addition compounds” is used. For chemical accuracy, I recommend using “Michael acceptor compounds” instead, as this terminology more accurately reflects their role as electrophilic species in the reaction. This change should be applied consistently throughout the text, including the manuscript title.
- Please include a brief description of the analytical methods used in the study, such as LC-MS and NMR, within the supporting information. Additionally, provide appropriate figure legends or explanatory notes for any relevant data presented in the Supporting Information.
- Please expand the figure legends for Figures 2–4 to include key experimental details such as sample volume, compound concentration, incubation time, and other relevant conditions. This additional information will help improve clarity and allow for better interpretation and reproducibility of the data.
- In the Discussion section, the authors propose that an intramolecular interaction between the ortho-hydroxyl group and the ketonic oxygen may underlie the reduced antisickling activity of MMA-07 compared to MMA-06. To support this hypothesis and enhance mechanistic insight, it is recommended that molecular docking studies, if applicable, be performed to visualize the potential intramolecular interaction and evaluate its influence on hemoglobin binding.
Minor comments:
- The captions for Schemes 1 and 2 should be placed directly after the schemes themselves and before the reaction conditions.
- I recommend using the format “MMA-06 to -09” instead of listing each compound individually (e.g., “MMA-06, MMA-07, MMA-08, and MMA-09”). See line 310 as an example.
- Please format "Na2SO4" with proper subscripts for the numbers to ensure correct chemical notation. Additionally, review and correct all chemical formulas throughout the manuscript where applicable (e.g., lines 423, 433, 462 etc.).
- Line 540: “J” is missing for the signal at 7.02 ppm.
- Line 637: Please correct the number of references.
- In the supporting information file, the affiliation for 1 is missing.
Round 2
Reviewer 2 Report
Comments and Suggestions for Authors
The authors have carefully and fully addressed every comment raised.
Good Luck